# Programmatic considerations and evidence gaps for chikungunya vaccine introduction in countries at risk of chikungunya outbreaks: Stakeholder analysis

**Megan Auzenbergs** [1]*, **Clara Maure**[2], **Hyolim Kang**[1], **Andrew Clark**[1], **Oliver Brady**[1], **Sushant Sahastrabuddhe**[2‡], **Kaja Abbas**[1,3‡]

**1** London School of Hygiene and Tropical Medicine, London, United Kingdom, **2** International Vaccine Institute, Seoul, South Korea, **3** School of Tropical Medicine and Global Health, Nagasaki University, Nagasaki, Japan

‡ These authors share senior authorship on this work.
* megan.auzenbergs@lshtm.ac.uk

**Data Availability Statement:** Participants consented to interviews understanding that the data collected would not be shared publicly. The

## Abstract

Chikungunya can have longstanding effects on health and quality of life. Alongside the recent approval of the world's first chikungunya vaccine by the US Food and Drug Administration in November 2023 and with new chikungunya vaccines in the pipeline, it is important to understand the perspectives of stakeholders before vaccine rollout. Our study aim is to identify key programmatic considerations and gaps in Evidence-to-Recommendation criteria for chikungunya vaccine introduction. We used purposive and snowball sampling to identify global, national, and subnational stakeholders from outbreak prone areas, including Latin America, Asia, and Africa. Semi-structured in-depth interviews were conducted and analysed using qualitative descriptive methods. We found that perspectives varied between tiers of stakeholders and geographies. Unknown disease burden, diagnostics, non-specific disease surveillance, undefined target populations for vaccination, and low disease prioritisation were critical challenges identified by stakeholders that need to be addressed to facilitate rolling out a chikungunya vaccine. Future investments should address these challenges to generate useful evidence for decision-making on new chikungunya vaccine introduction.

## Author summary

The first vaccine to prevent chikungunya fever has been recently approved in November 2023 by the US FDA and multiple chikungunya vaccine candidates are in different phases of the development pipeline. These will be the first-ever vaccines against an alphavirus and offer new technologies for vaccine development against other viruses of the same family that may cause future epidemics. We interviewed stakeholders from areas at risk of chikungunya outbreaks across Latin America, Asia and Africa, and identified gaps in Evidence-to-Recommendation criteria that should be addressed alongside vaccine introduction. Our findings show that stakeholders from different regions prioritised

data used in this analysis will be shared anonymously upon reasonable request. LSHTM (The London School of Hygiene & Tropical Medicine) ethics board approved this study. Interested readers can contact Chaelin Kim at chaelin.kim1@lshtm.ac.uk to access the data.

**Funding:** This project was funded by the International Vaccine Institute. HK is supported by the Vaccine Impact Modelling Consortium. OJB was supported by a UK Medical Research Council Career Development Award (MR/V031112/1). KA is supported by the Vaccine Impact Modelling Consortium (INV-034281) and the Japan Agency for Medical Research and Development (JP223fa627004). The study funder was involved in study design, data collection and analysis, decision to publish, and preparation of the manuscript.

**Competing interests:** The authors have declared that no competing interests exist.

chikungunya differently, but all stakeholders agreed that the unknown burden of disease, undefined target populations for vaccination and non-specific disease surveillance were challenges that needed to be addressed imminently. To address these gaps, the involvement of stakeholders in all phases of vaccine development and rollout will be crucial to uncover future challenges and to ensure vaccine equity.

## Introduction

Chikungunya is a mosquito-borne neglected tropical disease (NTD) caused by the chikungunya virus (CHIKV), an alphavirus spread by the mosquito vectors *Aedes aegypti* and *Aedes albopictus*. Symptoms associated with chikungunya fever are often mild, but can be associated with severe morbidities, such as persistent arthralgia, reported in 88% of cases up to one month after infection [1] and severe chronic arthralgia lasting years after infection [2, 3]. The severe, chronic morbidities associated with chikungunya fever can have longstanding effects on health and quality of life.

The stochastic transmission dynamics of CHIKV make it difficult to predict when the next outbreak will occur or if CHIKV will become endemic in any specific setting. Chikungunya cases have historically been clustered in tropical areas with warm, humid climates where the vectors thrive and cause recurring outbreaks of chikungunya fever. In a related systematic review and modelling study, we inferred subnational heterogeneity in the force of infection and transmission dynamics as well as identified both endemic and epidemic settings coexisting within countries such as Brazil, Ethiopia, and India [4]. However, the increasing spread of the vector to more geographic regions due to climate change poses a greater risk of CHIKV to more people in the future [5, 6]. CHIKV-carrying mosquitoes are currently endemic in the Americas, parts of Africa, and Southeast Asia [7]. These geographical regions are at high-risk of infection and carry the greatest burden of global chikungunya cases.

On November 9, 2023, the first ever chikungunya vaccine, Ixchiq, was approved by the US Food and Drug Administration (FDA) [8]. The vaccine was developed by Valneva, alongside investment from The Coalition for Epidemic Preparedness Innovations (CEPI) [9, 10]. The vaccine was approved for use in individuals 18 years and older who are at an increased risk of exposure to CHIKV. Currently the vaccine is a one-dose vaccine and is estimated to cost US $350 per dose for US travellers with a discounted cost of US$10–20 per dose in low- and middle-income countries [11]. Whilst chikungunya was not included in the Global Alliance for Vaccines and Immunisation (GAVI) Vaccine Investment Strategy for 2024, a learning agenda will be developed to identify the gaps which need to be addressed before such an investment can be considered. CEPI's support for the chikungunya vaccine development alongside GAVI's learning agenda [12] for this vaccine provides a pathway towards equitable access for chikungunya vaccines in countries at risk of chikungunya outbreaks [13]. A schematic showing the chikungunya vaccine development to introduction pathway is presented in Fig 1.

A chikungunya vaccine provides primary value in reducing global burden of CHIKV and long-term disabilities associated with CHIKV infection. It also provides additional significant value since this is the first vaccine against an alphavirus genus in the family *Togaviridae*, thereby enabling a novel vaccine development platform against emerging alphaviruses in the family *Togaviridae* [14]. As the risk of emerging infections increases with global travel and climate change, having an existing mechanism for developing a vaccine against an emerging pathogen expedites global outbreak response and vaccine development, as was done with mRNA and viral-vector vaccines during the COVID-19 pandemic [15].

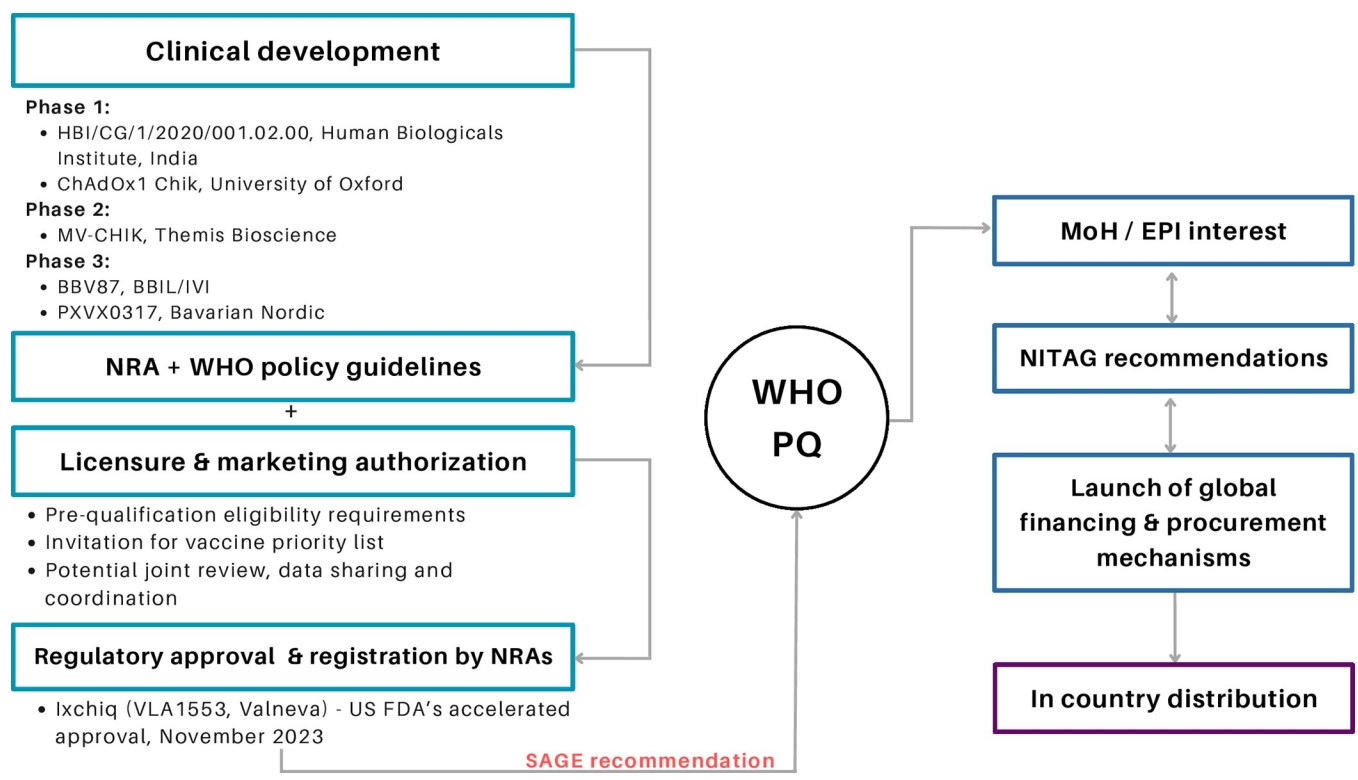

**Fig 1. Chikungunya vaccine development to introduction pathway.** Schematic showing the vaccine development process for the chikungunya vaccine alongside stages of licensure and evidence-based recommendations for policy making decisions. WHO–World Health Organization; MoH–Ministry of Health, NRA—National Regulatory Agencies, SAGE–Strategic Advisory Group of Experts, EPI–Expanded Program for Immunization, NITAG–National Immunization Technical Advisory Group, PQ- Pre-qualification.

The chikungunya vaccine value profile provided a high-level holistic assessment of available evidence to inform the potential public health, economic, and societal value of chikungunya vaccines in the development pipeline [16]. However, evidence to recommend chikungunya vaccine introduction is needed, including the disease burden, benefits and harms of chikungunya vaccination, values and preferences of the target population, acceptability to stakeholders, resources use and economic impact, equity, and feasibility [17]. As the global risk of CHIKV infection increases alongside introduction of the first chikungunya vaccine, we urgently need to understand the target populations for the new vaccine in addition to context specific social, logistical and financial barriers to rolling out the vaccine [18]. To date, qualitative research on chikungunya has been limited to patient experience, specifically quality of life and coping strategies following infection [19, 20]. Further, there is a lack of research exploring cultural explanations and conceptualizations of CHIKV aetiology in different geographical areas [21].

We aim to identify gaps in the Evidence-to-Recommendation (EtR) criteria needed to assess the introduction of chikungunya vaccine. To our knowledge, this is the first study to synthesise stakeholder perceptions on chikungunya outbreaks and vaccination by interviewing a diverse sample of global, national and subnational stakeholders involved in different elements of chikungunya epidemiology, policy, outbreak control and vaccinology. We provide timely implications for decision-making alongside qualitative data from a robust sample of stakeholders to inform introduction of the first available and licensed chikungunya vaccine.

## Methods

### Ethics statement

Ethical approval for this project was received from the London School of Hygiene and Tropical Medicine in January 2023, project reference number 28292. Written consent was obtained from all stakeholders ahead of the interviews.

### Stakeholder selection

We conducted a scoping review on chikungunya epidemiology to identify geographical regions at risk of chikungunya outbreaks alongside countries with ongoing clinical trials of chikungunya vaccine candidates. The regions of Latin America, Africa and Asia were prioritised for stakeholder identification. From here, a diverse list of contacts was created using purposive and snowball sampling of organisational databases, search engines and input from project coordinators at the International Vaccine Institute who oversee several clinical trial networks for chikungunya. At the end of all interviews, we requested stakeholders to recommend colleagues that would be also interested in taking part in an interview, to which a follow-up invite was sent. Further explanation of the stakeholder sampling framework can be found in S1 Fig.

Participants were first grouped into geographical categories and then grouped into one of three hierarchical categories: global, national or subnational stakeholders, referred to later as stakeholder tiers. From all geographical regions sampled, global stakeholders included experts from international organisations focused on immunisation and academics with a focus on arbovirus research in one of the aforementioned high-burden regions. National stakeholders included experts working at country-level ministries of health or within a policy sector for vaccine regulatory approval and oversight. Subnational stakeholders included clinicians, laboratory scientists and community health workers with experience working with chikungunya patients or in high-burden areas. Participants were geographically representative of chikungunya burden and evenly split across stakeholder tiers.

### Data collection

We developed a semi-structured interview questionnaire (S1 Table) through consultations with experts in vaccine epidemiology and reviewed existing studies evaluating the perception of stakeholders on other vaccine introductions and rollouts. Questions were focussed on perception of chikungunya outbreak risk, barriers to chikungunya vaccination, and pathways to advance the chikungunya vaccine agenda in the future. At the time of the interviews, there was no licensed chikungunya vaccine, although several vaccine candidates had ongoing or completed phase III clinical trials. As interviews were conducted, questions were revised to reflect new topics that emerged. Biweekly meetings with the research team occurred to ensure the interviews were going smoothly and new themes that emerged through data collection were discussed.

### Data analysis

We conducted qualitative semi-structured interviews via video call during which detailed notes were transcribed. We analysed the interview data through an iterative process using MAXQDA 2022 (VERBI Software, 2021) for data analysis and codebook development was done following the methods discussed in MacQueen et al.[22]. We use inductive and deductive coding to analyse the raw interview data. We categorised the coded data into themes. Thematic

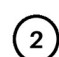
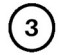
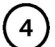
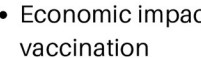
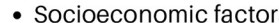
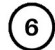
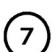

**FRAMING THE PROBLEM**
1
- Burden of disease
- Clinical characteristics
- Regional differences

**BENEFITS & HARMS OF VACCINATION**
2
- Efficacy and safety
- Indirect effects

**VALUES & PREFERENCES OF TARGET POPULATION**
3
- Well-defined target population
- Perception of the disease
- Perception of vaccination
- Subgroup differences

**ACCEPTABILITY TO STAKEHOLDERS**
4
- Acceptability of vaccination
- Financial and ethical considerations

**RESOURCES**
5
- Economic impact of vaccination
- Socioeconomic factors

**VACCINE EQUITY**
6
- Access to vaccination
- Stigma

**FEASIBILITY**
7
- Storage and distribution of vaccines
- Vaccine availability
- Information management
- Disease surveillance

**Fig 2. Evidence-to-Recommendation criteria for chikungunya vaccine introduction.** The Evidence-to-Recommendation criteria for chikungunya vaccine introduction is based on the World Health Organization's Guidance on an adapted Evidence-to-Recommendation Process for National Immunization Technical Advisory Groups.

differences between geographical regions were first identified and then stakeholder tiers were analysed.

## Identification of evidence gaps

Guidance on the EtR used by national immunization technical advisory groups (NITAGs) [17] was used to identify evidence gaps in current chikungunya knowledge and research, as shown in Fig 2. EtR criteria were then aligned with stakeholder perspectives and grouped by geographical region to highlight regional evidence gaps.

## Results

### Participant characteristics

Between January-February 2023, approximately 60 stakeholders were emailed and invited to take part in an interview. Overall, a total of 18 stakeholder interviews were conducted via video call between February and July 2023 (see Table 1).

### Implications for decision making

We identified several themes for challenges associated with chikungunya vaccine introduction. Notable differences exist within stakeholders in different organisation tiers and by geographical regions (see Table 2).

**Table 1. Participant characteristics by geographical region and type of stakeholder.**

| Geographical region | Country | Type of stakeholder | Number of interviewees |
|---|---|---|---|
| Latin America | Brazil | National | 3 |
| | | Subnational | 1 |
| | Guatemala | Subnational | 1 |
| | Colombia | National | 2 |
| | | Subnational | 2 |
| Asia | Thailand | National | 2 |
| | India | National | 1 |
| | | Subnational | 1 |
| Africa | Kenya | Subnational | 2 |
| International | | Global | 3 |
| | | Total | 18 |

## Unknown burden of disease

The disease burden of chikungunya is unknown in many settings, which was the most frequently mentioned barrier to uptake of a chikungunya vaccine reiterated by stakeholders across all organisation tiers and geographical regions, alongside awareness of the unpredictability of chikungunya outbreaks.

Stakeholders partially attribute the unknown burden to non-specific or insensitive surveillance as surveillance for CHIKV is often done alongside other arboviral diseases, such as dengue and Zika. Surveillance for CHIKV is also often based on clinical cases, so passive surveillance only, which stakeholders believe results in under-reporting as surveillance systems usually only capture the cases that seek medical attention. Because this type of case detection relies on symptomatic patients reporting to health systems, passive surveillance excludes less severe or asymptomatic infections.

Without a comprehensive understanding of disease burden, quantifying the economic burden, including direct and indirect costs of acute but also chronic symptoms, is difficult. This barrier primarily affected national and global level stakeholders as countries cannot advocate for interventions, such as vaccination, without a cost-effectiveness and risk benefit analysis.

> *"We need to better understand the burden of disease, disability adjusted life years (DALYs) lost, the benefits of a vaccine in terms of reducing morbidity and work loss."*
>
> *-Programme lead for chikungunya, international organisation*

> *"We don't have good chikungunya surveillance, and it is usually paired with surveillance for dengue and Zika. Surveillance can be coupled with dengue and Zika, but a good surveillance system should look for mild cases of chikungunya, not just severe cases that will look for medical attention. A lot of mild chikungunya cases are not found or not reported"*
>
> *-Paediatric infectious disease specialist, Guatemala*

## Geographical variations in disease burden

We observed regional differences in stakeholder perceptions around chikungunya burden. For African countries, other febrile illnesses make fevers associated with chikungunya difficult to accurately detect.

**Table 2. Themes and challenges presented by stakeholders in different organisation tiers and by geographical regions.**

| Theme | Challenges presented by stakeholders in different organisation tiers | | | Challenges presented by stakeholders in different geographical regions | | | |
|---|---|---|---|---|---|---|---|
| | Subnational level | National level | Global level | Latin America | Asia | Africa | International |
| **Unknown burden of disease** | A lack of diagnostic sensitivity and laboratory capacity in the most affected areas results in under diagnosis and under reporting of chikungunya | Surveillance for chikungunya is lacking in areas, which makes it difficult to understand which areas are most affected, detect outbreaks and respond accordingly | Without a good understanding of disease burden, demonstrating the economic burden of chikungunya or economic impact of a vaccine is challenging | Non-specific disease surveillance makes it difficult to distinguish the burden of disease between chikungunya, dengue and Zika | Unknown chikungunya burden makes it difficult to advocate for chikungunya prevention and vaccination over dengue | Inability to detect actual chikungunya cases amongst other febrile illnesses, such as malaria, results in a large under-estimation in disease burden | Prioritisation of the vaccine in certain geographical regions is uncertain, making investment case for the vaccine difficult |
| **Chikungunya has a high burden of morbidity, but not mortality making disease prioritisation uncertain** | Public perception around chikungunya can be lacking in areas with endemic dengue circulating | It is difficult to prioritise chikungunya over other pathogens (specifically dengue or Zika) when it comes to investing in developing improved laboratory and surveillance methods | Country buy-in is important for future vaccine investment strategies | Despite co-circulation of chikungunya with other arboviruses and lower mortality rates, the chikungunya vaccine is a priority and countries are preparing for vaccine rollout | Lack of buy in from national vaccine policymakers to prioritise the chikungunya vaccine over the dengue vaccine | Prioritisation of other diseases with higher mortality rates means chikungunya is rarely discussed, and public awareness about the disease is lacking | Varying levels of prioritisation and support for the vaccine makes it difficult to plan for vaccine introduction |
| **Target population for the chikungunya vaccine is not well defined** | Vaccine confidence and public perception of a chikungunya vaccine would affect the success of a vaccine roll-out | Ensuring that the right infrastructure is in place to deliver the vaccine is difficult because the exact target population and delivery method (outbreak response or routine immunisation) is unknown | Understanding the exact use of the vaccine and the target populations are important for stockpile estimates, which are part of a global vaccine investment strategy | Anticipated use in outbreak response and affected areas, but approval of the current vaccine only for use in 18-years and older individuals means uncertainty if/ when children can be vaccinated | Disease burden varies greatly within some countries, so subnational infrastructures would need to be in place to improve diagnostics and support vaccination at the local level | Research shows a high burden amongst children, but lack of age-specific seroprevalence data makes it hard to define a target population, the vaccine has also not been evaluated in children | Following the safety approvals for the vaccine and recommended age groups may make outbreak trajectory uncertain if outbreak data shows high burden amongst children |
| **Chikungunya has specific climate or vector factors to consider** | Different disease burdens are experienced by different sub-populations because of vector exposure | As vector epidemiology changes, chikungunya may become endemic in some countries, this has implications for vaccine stockpiles and roll-out | The technology behind the chikungunya vaccine may aid vaccine development of other alphaviruses in the *Togaviridae* family | Chikungunya and dengue cocirculate and concurrent outbreaks have occurred. It is important to understand how to deploy both the dengue and chikungunya vaccines in outbreak settings | Vector viability can differ within the same country, so sometimes local prevention measures and vaccination would be preferred over national programmes or campaigns | The animal reservoir in Africa demonstrates sylvatic transmission and viral evolution, so global chikungunya prevention should be concerned with natural origins of the virus | As global travel patterns and climate change affect viability of settings for the chikungunya vector, epidemic trends and spatial epidemiology may shift |

*"The burden of disease is not well defined for chikungunya. We do not understand the nature of outbreaks and the burden of disease in African nations because it is hidden in other febrile illnesses (malaria, etc.)."*

*-Programme lead for chikungunya, international organisation*

In South America, stakeholders attribute unknown burden mostly to passive surveillance and the fact that outbreaks of chikungunya and dengue sometime occur concurrently.

*"There is lots of under-reporting because current chikungunya surveillance is based on clinical cases, passive surveillance only, not active case detection, so we are only capturing cases that seek medical help, not community level cases."*

-*Neglected tropical diseases division, national organisation, Brazil*

In India, the subnational burden of disease is of concern if a chikungunya vaccine were to be rolled out. This is of particular importance given the large population size of India and infrastructure needed to manufacture enough vaccine doses and deliver these doses to many people.

*"There are subnational variations in burden within India. It is a question if the vaccine would be rolled out as a pan-national vaccine, or if it would be like the Japanese Encephalitis vaccine, which is restricted to only a few areas. Burden is limited in some parts of India, except for big states, like Delhi, Uttar Pradesh."*

-*Infectious disease clinician, India*

### Diagnostics

Subnational stakeholders identify additional concerns around improving laboratory diagnostics and not overloading laboratory capacity alongside vaccine interventions. At the subnational level, stakeholders expressed concerns on diagnostic sensitivity and capacity since some diagnostic tests are unable to detect CHIKV infections early enough, and that many high-risk areas are not well-equipped with the laboratory equipment required for CHIKV diagnosis and samples therefore are shipped elsewhere [23, 24].

*"The amazon region of Brazil does not have good laboratory capacity for diagnostics, most samples are shipped to São Paulo, so most local diagnoses are left to clinical diagnoses. Enhanced serological testing for burden estimates would not be possible because of the remoteness of the area."*

-*Nurse and laboratory specialist, Amazon region, Brazil*

*"You need to know both the symptomatic and asymptomatic burden of chikungunya—we will need to determine the asymptomatic burden and confirm what genotypes are circulating. To do this, we require: good serology kits for IgM and IgG, good PCR that will detect the different circulating strains and a well working lab team."*

-*Virology laboratory specialist, India*

### Unpredictable outbreaks

In places where chikungunya is not endemic, outbreaks are unpredictable. The unpredictability of outbreaks affects stockpiling of vaccine as it is difficult to estimate the time, duration and number affected during each outbreak.

*"Chikungunya tends to come in waves, but not predictable waves within regions and countries, the prioritisation and national public health interest (and therefore funding) is low because the waves only come once in a while and are unpredictable, so it is difficult to maintain attention to efforts."*

-*Programme lead for chikungunya, international organisation*

In other regions, policy stakeholders believe chikungunya would have to become an endemic disease for priority to be given to the chikungunya vaccine.

*"Even if we have an efficacious [chikungunya] vaccine, there are a number of challenges for public use. In comparison to dengue, chikungunya would need to be endemic in our region [Southeast Asia]. However, from the epidemiology, we see the number of chikungunya cases occurring per year are less than dengue."*

*-Vaccine policy & safety, national organisation, Thailand*

## Prioritisation of chikungunya over other arboviral diseases

Because chikungunya has a high burden of morbidity, but not mortality in many regions, stakeholders admit it is currently not a high-priority disease. Since countries with the greatest burden of chikungunya also have high burdens of other arboviral diseases, specifically dengue, there are often competing priorities. For example, stakeholders in India and Southeast Asia mostly prioritised chikungunya lower than dengue. One stakeholder from Thailand even refused to partake in an interview because they saw the promotion of a chikungunya vaccine to detract from resources being allocated to dengue vaccine roll-out. Despite this perceived competition, stakeholders in Latin and South America overall had the greatest interest in the chikungunya vaccine and were confident in vaccine roll-out despite concurrent dengue out-breaks, albeit with some concerns about public perception of the vaccine due to lack of per-ceived risk of morbidities associated with chikungunya.

*"Now that the dengue vaccine is about to be licenced globally in the very near future, dengue may be a higher priority in the same countries where chikungunya is also a problem, so den-gue will probably be ahead of chikungunya in the priority list. Latin America is the region that has the most interest in chikungunya, the most concern and high prioritisation. Certain sectors in India may be interested, but overall, broadly, prioritisation in India is lower, they will prioritise dengue over chikungunya"*

*-Programme lead for chikungunya, international organisation*

*"The size of morbidity and mortality is lower for chikungunya than dengue, moreover, we rarely see a mortality rate from chikungunya that is similar to that of dengue, particularly in children, so severity (DALYs) is less, burden is less than dengue. The number of cases of chi-kungunya does not ring a bell and there is a longer duration between outbreaks, which is hard to predict. We [Thailand] have a number of competitive health problems and we have many things on the priority list".*

*-Vaccine policy & safety, national organisation, Thailand*

*"There will be challenges rolling out the chikungunya vaccine. On an individual level, people won't see it as an important vaccine because there is a feeling that chikungunya is a mild dis-ease. The perception is that it is not as important as other diseases. People won't be as eager to get the vaccine, which is different from dengue. Lots of people have seen severe dengue, so if you see people in the hospital with dengue, you know it is a severe disease. But with chikungu-nya, people usually don't go to the hospital and if they do, they usually don't die from chikungunya."*

*-Paediatric infectious disease specialist, Guatemala*

Stakeholders state that the lack of vaccine evaluation and an underinvestment in chikungunya research in Africa perpetuates vaccine inequity. Whilst some stakeholders in Latin America have a better understanding of how a chikungunya vaccine would be rolled out, other stakeholders in Africa are concerned about the lack of knowledge about the disease in their geographical region.

*"Chikungunya was discovered in Tanzania, there has never been a vaccine trial anywhere in Africa, there are no discussions about vaccine evaluations in Africa. And yet suddenly, we have all these advanced programmes for chikungunya and the chikungunya pipeline is very healthy, but none of those products actually have a strategy for evaluation in Africa, as far as I am aware . . .. This promotes inequity."*

*-Vaccinologist and One Health expert, Kenya*

Because of prioritisation of other diseases, the community engagement process to raise awareness about chikungunya is lacking. This concern is emphasised by African stakeholders, where the lack of awareness about the disease poses challenges for future interventions and highlights evidence gaps associated with perception of the disease and stigma.

*"Chikungunya is known, but it is stereotyped as a disease that came from spirits from the ocean, or is linked to witchcraft, they think that a treatment is drinking boiled papaya leaves. People know this thing (chikungunya) exists, but general knowledge about chikungunya is lacking. Most people know about malaria, what symptoms are, they know about malaria treatment, but if you talk about chikungunya they laugh at you because public engagement has not been done for chikungunya."*

*-Academic researcher, Kenya*

## Target population for the chikungunya vaccine is not well defined

The target population for a vaccine includes individuals within defined demographics and geographies that are eligible for a vaccine intervention. The unknown target population for the chikungunya vaccine presents challenges to all tiers of stakeholders. For global stakeholders, the biggest implication for unknown target population is the impact this has on industry manufacturing.

*"Things have already been done and spearheaded by individual vaccine manufacturers to push the [chikungunya vaccine] development path forward. The biggest impediment for chikungunya vaccine development has not been technical, but defining what the market is for a chikungunya vaccine. There is not a huge amount of public funding for chikungunya vaccine development because of morbidity and mortality and burden issues. The commercial market is limited. Big players are not vaccine multinationals, but more intermediate developers, or ones located in endemic countries . . . the biggest impediment has been what is the commercial market that makes it worth developing?"*

*-Programme lead for chikungunya, international organisation*

For national and subnational stakeholders, the undefined target population presents more concerns for vaccine roll-out logistics in their countries.

*"What we see in the Paraguay outbreak is that more children have been infected and there have been more fatalities in children, so we need more information on this. The diagnostics and surveillance previously available have been weak to detect the type of infection some of these children have . . . Maybe children were not affected so much in previous outbreaks, but going forward we need to be aware of child infections. For example, from which age should we vaccinate?"*

*-Physician and clinical researcher, Colombia*

Additionally, concerns around vaccine hesitancy and vaccine equity highlight the need for identification of target populations ahead of vaccine roll-out and ensuring that vaccine roll-out is packaged alongside advocacy campaigns.

*"Some important questions that need to be addressed soon, include: if vaccine supplies are limited, which population groups are considered priority groups? The purpose of vaccination is to achieve what objective? To ensure equity in vaccine distribution, what steps do we need to take?"*

*-Neglected tropical diseases division, national organisation, Brazil*

## Climate sensitivity of Chikungunya vectors

Sub-populations experience different disease burdens because of differences in exposure to mosquito bites. Stakeholders explain that variations in vector exposure is an important consideration in vaccine roll-out because certain populations are disproportionately affected. As climate change impacts vector and virus population dynamics, CHIKV may become endemic in some countries, this has implications for vaccine stockpiles and roll-out.

*"In Guatemala, we have a lot of areas of high rain, humidity and areas for mosquitoes to grow. Alongside, our population is growing, and we have a large urban population, so more people in small places, which makes the perfect conditions for an outbreak."*

*-Paediatric infectious disease specialist, Guatemala*

*"Africa is the only continent that has reported sylvatic circulation, between primates and mosquitoes. This is the natural reservoir for chikungunya virus, so we need to do a thorough study of chikungunya in Africa because even if the vaccine is rolled out elsewhere and people are protected, we don't know what strains will come again from the natural habitat for the virus, especially as climate changes the evolution of viruses. So, these viruses, as much as you can control them elsewhere, these viruses will again spread from their original source. So, to address challenges for chikungunya, it is better to address them from the source."*

*–Academic researcher, Kenya*

## Existing evidence gaps

Regional stakeholder perspectives were aligned with EtR criteria to highlight existing gaps in knowledge. Where stakeholders believed an EtR criterion was a current challenge or a gap in knowledge, it was recorded in Table 3. If no annotation was made for any of the criterion, this means the topic was not discussed during the stakeholder interviews or the topic was not identified as a current evidence gap.

**Table 3. Mapping stakeholder perspectives with Evidence-to-Recommendation criteria.** Stakeholder perspectives were aligned with Evidence-to-Recommendation criteria to highlight existing gaps in knowledge. Cells in the table without data mean these topics were not discussed by any of the stakeholders.

| Evidence-to-Recommendation Criteria | | Stakeholders identifying a specific criterion as a knowledge gap or challenge | | | | Implications |
|---|---|---|---|---|---|---|
| | | Africa (n = 2) | Asia (n = 4) | Latin America (n = 9) | Global (n = 3) | |
| 1. Framing the problem | Burden of disease | 2/2 | 4/4 | 7/9 | 2/3 | Unknown burden of disease leads to unknown epidemiology and lower prioritisation of the disease |
| | Clinical characteristics | 2/2 | | 4/9 | | Misdiagnosed cases result in under-reporting |
| 2. Benefits and harms of vaccination | Vaccine efficacy and safety | 2/2 | 3/4 | 1/9 | 2/3 | Vaccine efficacy is based on neutralizing antibodies as a potential immune correlate of protection against chikungunya infection. The first chikungunya vaccine (VLA1553) to get regulatory approval has demonstrated good immunogenicity profile. |
| | Indirect effects of vaccination | 1/2 | | | | Broader benefits include chikungunya vaccine serving as a prototype vaccine for other alphaviruses |
| 3. Values and preferences of target population | Well-defined target population | 2/2 | 3/4 | 5/9 | 2/3 | Unknown burden leads to unknown target populations and risk areas |
| | Perception of the disease | 1/2 | 2/4 | 4/9 | 1/3 | Variable awareness about the disease can affect public perception of risk |
| | Perception of vaccination | 1/2 | 3/4 | 6/9 | | Variable risk perception can affect willingness to get vaccinated |
| | Differences in subgroups | 2/2 | 2/4 | 8/9 | 1/3 | The vector and disease affect certain subgroups disproportionately |
| 4. Acceptability of the vaccine | Financial & ethical considerations | 2/2 | 1/4 | 3/9 | | Government prioritisation of disease may affect willingness to pay for the vaccine versus private market availability in country |
| 5. Resources | Economic impact of vaccination | 1/2 | 2/4 | 1/9 | 3/3 | Unknown disease burden leads to evidence gaps in economic evaluation |
| | Socioeconomic factors | | 4/9 | | | People living in poverty are disproportionately affected |
| | Diagnostics and laboratory capacities | 2/2 | 2/4 | 4/9 | 1/3 | Non-specific diagnostics and non-detection of asymptomatic infections leads to misreporting and underreporting |
| 6. Vaccine equity | Access to vaccination | 1/2 | | 6/9 | | If the vaccine is not available without out-of-pocket charges, those affected by the disease may not be able to or willing to pay privately |
| | Stigma | 1/2 | | | | Lack of public communication about the disease leads to stigmatising misconceptions |
| 7. Feasibility | Storage and distribution | 1/2 | 2/4 | 6/9 | 2/3 | Understanding how the vaccine will be delivered (i.e. outbreak response) is important |
| | Vaccine availability | 1/2 | 1/4 | 1/9 | | If the vaccine will be used in outbreak response, considerations for vaccine stockpile are necessary |
| | Information management | 2/2 | 2/4 | 6/9 | 1/3 | The vaccine should be rolled-out in parallel to a communications package to raise awareness about both the disease and vaccine |
| | Disease surveillance | 2/2 | 2/4 | 4/9 | 1/3 | Non-specific disease surveillance can result in under-reporting or unknown disease burden |

## Discussion

We infer from our stakeholder analysis that unknown disease burden, diagnostics, non-specific disease surveillance, undefined target populations for vaccination, and low disease prioritisation are critical challenges that need to be addressed to facilitate rolling out a chikungunya vaccine. Future investments should address these challenges to generate useful evidence for decision-making on new chikungunya vaccine introduction.

Both disease burden and surveillance were highlighted as gaps in the Evidence-to-Recommendation criteria across all geographical regions, further stressing these as major issues that need addressing ahead of vaccine rollout. Paucity of data and research illustrating the

disease burden of chikungunya, exacerbated by non-specific disease surveillance presents several challenges. The disease burden of chikungunya remains unknown and likely underestimated in many high burden settings due to a lack of chikungunya-specific disease surveillance [25]. Laboratory capacity and existing diagnostics for detecting chikungunya infection are limited in some high burden settings [26]. Passive surveillance methods currently used in many settings only pick up clinical cases of chikungunya presenting to hospital, resulting in an under diagnosis of asymptomatic and less severe infections. Analysis of age-stratified seroprevalence data is a useful method for estimating long-term average infection burden [4]. In some African settings, misdiagnosis of chikungunya as another febrile illness, such as malaria, is common, which is concerning, given that research shows a higher burden of chikungunya in children [27]. Accurate detection and surveillance of alphaviruses in vectors is especially important in Africa (and other malaria endemic areas) where existing zoonotic reservoirs exist and there has been an increasing frequency of chikungunya detection in recent years [28]. Accordingly, stakeholders in Africa highlighted the indirect effects of vaccination as a current evidence gap that would be valuable to address alongside cross-protections from chikungunya and other viruses.

The unknown disease burden also affects prioritisation of chikungunya, both in terms of national vaccine policy decisions [25] and public perception of chikungunya risk [21]. Several stakeholders highlighted that by focusing on chikungunya vaccination, resources are taken away from dengue vaccination and prevention, which many stakeholders, especially those in South and Southeast Asia, believe is a higher priority on country agendas. In contrast, stakeholders in Latin America affirmed a higher prioritisation of chikungunya in national vaccine policy agendas, but voiced concerns that public perception of chikungunya risk was skewed by a greater awareness about dengue, including symptoms, transmission and infection risk. Because chikungunya is often seen as a disease with low mortality, stakeholders voiced concerns in the public perception of risk [21,29]. Lower prioritisation of chikungunya is concerning because the long-term chronic disability of chikungunya such, as arthritis, can be debilitating, putting stress on health care systems and diminishing economic productivity. These health deficits for chikungunya are not usually captured in global health assessments despite the large populations currently at risk [30]. By illustrating the true burden of chikungunya and its long-term health and economic impact, stakeholders were confident that prioritisation and public perception of chikungunya risk can be increased.

Concern for social factors affecting vaccine rollout were varied across geographical regions. Stakeholders in Africa and Latin America identified vaccine perception and hesitancy, information management and socioeconomic factors affecting vaccine uptake as current challenges more often than stakeholders in Asia. This could be attributed to the overall perception and prioritisation of chikungunya—stakeholders in Latin America saw chikungunya as a higher priority disease whilst stakeholders in Asia stated other diseases with competing interests were a higher priority. This differential prioritisation could affect concern for social factors around vaccination, showing a greater level of thought has been put into chikungunya vaccine equity amongst stakeholders that see the vaccine as more favourable. Stakeholders discussed synergies with other vaccination programmes, specifically citing lessons learned from distribution and administration of COVID-19 vaccines. Stakeholders in Latin America and Asia believed lessons learned during the COVID-19 pandemic could be leveraged for the chikungunya vaccine, however, stakeholders in Africa saw the ongoing inequity of COVID-19 vaccines in Africa to perpetuate concerns about chikungunya vaccine equity in the African continent.

New chikungunya vaccines provide broader value beyond the direct benefits of lowering the chikungunya disease burden. These will be the first-ever vaccines against an alphavirus and thereby offer new platforms for vaccine development against other alphaviruses of the

family *Togaviridae* that may emerge to cause epidemics and potential for pandemics. Further, the lessons learned, and technologies developed by the chikungunya vaccine will pave the way for new regulatory approval processes as vaccines can be approved based on immunogenicity data estimated by measures of neutralizing antibodies as a potential immune correlates of protection, instead of vaccine efficacy estimates based on disease events [31].

Our study has limitations. By limiting our analysis to stakeholders in regions at risk of chikungunya outbreaks, stakeholders in regions at future risk of chikungunya invasion due to climate change were excluded from our interview sample. We used a limited number of organisational databases that so possibly not all relevant stakeholders were identified. Despite contacting over 60 stakeholders, the response rate was low, especially in Africa. When stakeholders were referred by other stakeholders, they were more likely to participate, suggesting that use of purposive sampling in addition to the low response rate could result in selection bias. The sample of interviewees is geographically representative of current willingness to roll-out the chikungunya vaccine, but the number of participants by region is not necessarily proportional to disease burden. For example, Latin America had the greatest number of participants across all geographies and it was also the region with the most eagerness to rollout the vaccine; however, the burden of disease in Africa is estimated to be relatively high, especially in children [27], and this burden was not proportional to the sample size of stakeholders from the African region included in our study. This limited sample size for Africa could be attributed to topics mentioned in interviews with stakeholders who expressed concern that chikungunya epidemiology is not currently well documented and disease awareness is low across the African region. Despite the low sample size, we were still able to interview stakeholders from three different high burden geographical regions, six different countries, and across the international, national and subnational organisation tiers, lending to diverse perspectives that will be valuable in making future decisions about chikungunya vaccine introduction and delivery strategies.

Especially given shifts in global travel patterns, urbanisation and climate change, as vector viability changes, public health officials must collaborate to improve surveillance, prevention, and control programmes for arboviral diseases [32–34]. In July 2023, the European Centre for Disease Prevention and Control (ECDC) announced an increasing risk of mosquito-borne disease in Europe following the spread of *Aedes* mosquito species capable of transmitting CHIKV [35]. While our analysis focused on perspectives in regions at current risk of chikungunya outbreaks and excluded Europe, the rising concern of transmission-competent mosquito populations in Europe highlights just one aspect of how changing climate patterns can shift the future epidemiology of chikungunya outbreaks. To address these evolving patterns, the involvement of stakeholders in all phases of vaccine development and rollout alongside risk assessment and climate sensitivity of chikungunya will be crucial to uncover challenges and gaps to be addressed in the future.

## Supporting information

**S1 Fig. Sampling framework used to identify individuals invited for the stakeholder interviews.** First, articles on chikungunya vaccination were identified from a PubMed search, limited to geographical regions with a high risk of chikungunya outbreaks. There was no language restriction on articles, but the publication date was limited to the last five years, in line with chikungunya vaccine development. Next, a database of individuals involved in chikungunya vaccine clinical trials were added to the list of stakeholders to contact. Third, WHO regional websites were used to identify individuals working on chikungunya in the AFRO, PAHO and SEARO regions. Individuals identified in steps 1–3 were then invited to partake in a

stakeholder interview. If successful contact was made, during the interview stakeholders were invited to refer colleagues that would also be interested in sharing perspectives.
(TIF)

**S1 Table. Interview questionnaire used in the semi-structured interviews.** These questions were used to guide the interviews, but participants were invited to speak freely and structure the interview how they saw best.
(DOCX)

## Acknowledgments

We would like to thank Asha Mary Abraham, Ashish Bavdekar, Doris Nyamwaya, Eolo Morandi Jr, Jacqueline Borin, Jamille Dombrowski, José Moreira, Elsa Marina Rojas Garrido, Maria Isabel Estupiñan Cárdenas, Mario Melgar, Myriam Tatiana Medina Bernal and Timothy Endy for participating in an interview and their valuable insights and perspectives shared.

## Author Contributions

**Conceptualization:** Megan Auzenbergs, Clara Maure, Kaja Abbas.

**Data curation:** Megan Auzenbergs, Clara Maure.

**Formal analysis:** Megan Auzenbergs, Clara Maure.

**Funding acquisition:** Clara Maure, Sushant Sahastrabuddhe.

**Investigation:** Megan Auzenbergs.

**Methodology:** Megan Auzenbergs, Clara Maure, Kaja Abbas.

**Project administration:** Megan Auzenbergs, Sushant Sahastrabuddhe, Kaja Abbas.

**Software:** Megan Auzenbergs.

**Supervision:** Kaja Abbas.

**Validation:** Megan Auzenbergs, Clara Maure, Hyolim Kang, Kaja Abbas.

**Visualization:** Megan Auzenbergs, Clara Maure, Andrew Clark, Oliver Brady.

**Writing – original draft:** Megan Auzenbergs, Clara Maure.

**Writing – review & editing:** Megan Auzenbergs, Hyolim Kang, Andrew Clark, Oliver Brady, Sushant Sahastrabuddhe, Kaja Abbas.

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
