## [Decision Letter · Decision Letter 0]

17 Jan 2024

Dear Ms Auzenbergs,

Thank you very much for submitting your manuscript "Programmatic considerations and evidence gaps for chikungunya vaccine introduction in countries at risk of chikungunya outbreaks: stakeholder analysis" for consideration at PLOS Neglected Tropical Diseases. As with all papers reviewed by the journal, your manuscript was reviewed by members of the editorial board and by several independent reviewers. The reviewers appreciated the attention to an important topic. Based on the reviews, we are likely to accept this manuscript for publication, providing that you modify the manuscript according to the review recommendations.

Sincerely,

Adly M.M. Abd-Alla, Prof asso.

Academic Editor

Abdallah Samy

Section Editor

Reviewer's Responses to Questions

**Key Review Criteria Required for Acceptance?**

**Methods**

-Are the objectives of the study clearly articulated with a clear testable hypothesis stated?

-Is the study design appropriate to address the stated objectives?

-Is the population clearly described and appropriate for the hypothesis being tested?

-Is the sample size sufficient to ensure adequate power to address the hypothesis being tested?

-Were correct statistical analysis used to support conclusions?

-Are there concerns about ethical or regulatory requirements being met?

Reviewer #1: The manuscript is based on interviews with stakeholders. The criteria mentioned above dont apply, however the considerations are important for CHIKV vaccine application.

Reviewer #2: The vaccines, like the cost or the manufacture, for the CHIKV should be included in the manuscript. And, is it possible to increase the stakeholders in Arfica?

**Results**

-Does the analysis presented match the analysis plan?

-Are the results clearly and completely presented?

-Are the figures (Tables, Images) of sufficient quality for clarity?

Reviewer #1: The results are explained in the text and clearly summarized in Tables.

Reviewer #2: The results clearly and completely presented.

**Conclusions**

-Are the conclusions supported by the data presented?

-Are the limitations of analysis clearly described?

-Do the authors discuss how these data can be helpful to advance our understanding of the topic under study?

-Is public health relevance addressed?

Reviewer #1: The conclusions are clearly presented and show novel geographic and stakeholder differences. This summary is important for the introduction of the currently available CHIKV vaccine.

Reviewer #2: The limitations of analysis are clearly described.

**Editorial and Data Presentation Modifications?**

Reviewer #1: - The paragraph on the Valneva vaccine (lines 45-56) needs to be updated since the vaccine got a FDA approval.

- Figures need a list of abbreviations

Reviewer #2: (No Response)

**Summary and General Comments**

Reviewer #1: Although this manuscript is not a typical research paper, it contains important considerations for CHIKV vaccine development and application.

Reviewer #2: Rolling out a chikungunya vaccine is critical issue in the future, especially for global travels are busy after the COVID-19, urbanization and climate change, and vector viability changes. This manuscript make a survey for the stakeholders among Asia, Latin American and Africa and discuss the limitation of the studies. This studies will be benefit for the Vaccine policy decision makers for the launch of the CHIKV vaccines.

PLOS authors have the option to publish the peer review history of their article (what does this mean?). If published, this will include your full peer review and any attached files.

Reviewer #1: No

Reviewer #2: No

Figure Files:

Data Requirements:

Reproducibility:

References

---

## [Decision Letter · Decision Letter 1]

15 Mar 2024

Dear Ms Auzenbergs,

We are pleased to inform you that your manuscript 'Programmatic considerations and evidence gaps for chikungunya vaccine introduction in countries at risk of chikungunya outbreaks: stakeholder analysis' has been provisionally accepted for publication in PLOS Neglected Tropical Diseases.

Best regards,

Adly M.M. Abd-Alla, Prof asso.

Academic Editor

Abdallah Samy

Section Editor

---

## [Editor Report · Acceptance letter]

28 Mar 2024

Dear Ms Auzenbergs,

We are delighted to inform you that your manuscript, "Programmatic considerations and evidence gaps for chikungunya vaccine introduction in countries at risk of chikungunya outbreaks: stakeholder analysis," has been formally accepted for publication in PLOS Neglected Tropical Diseases.

Best regards,

Shaden Kamhawi

co-Editor-in-Chief

Paul Brindley

co-Editor-in-Chief
